# The Predictive Value of Eosinophil Indices for Major Cardiovascular Events in Patients with Acute Decompensated HFrEF

**DOI:** 10.3390/medicina58101455

**Published:** 2022-10-15

**Authors:** Aslı Vural, Ertan Aydın

**Affiliations:** Clinic of Cardiology, Faculty of Medicine, Giresun University, Giresun 28200, Turkey

**Keywords:** eosinophil indices, acute decompensated heart failure with reduced ejection fraction, mortality

## Abstract

*Background and Objectives:* Heart failure is a chronic disease with a high risk of mortality and morbidity. In these patients, inflammatory markers have been shown to be associated with cardiovascular adverse outcomes and disease progression. To investigate the relationships between eosinophil indices and major cardiovascular events (MACE) in patients with acute decompensated heart failure (ADHF) with reduced ejection fraction. *Materials and Methods:* A total of 395 consecutive patients admitted to the intensive care unit (ICU) with ADHF and reduced ejection fraction between January 2017 and December 2021 were enrolled in this retrospective study. MACE was defined as the composite of death and re-hospitalization for ADHF within 6 months of index hospitalization. All-cause mortality and MACE were assessed with respect to relationships with eosinophil indices, including neutrophil-to-eosinophil ratio (NER), leukocyte-to-eosinophil ratio (LER), eosinophil-to-lymphocyte ratio (ELR), and eosinophil-to-monocyte ratio (EMR). *Results:* NER and LER were significantly higher in subjects with MACE. Absolute eosinophil, lymphocyte and basophil count, hemoglobin, serum Na^+^, albumin, and CRP, and EMR and ELR were significantly lower in subjects with MACE compared to those without. NT-proBNP (OR: 1.682, 95% CI: 1.106–2.312, *p* = 0.001), Na^+^ (OR: 0.932, 95% CI: 0.897–0.969, *p* < 0.001), NER (OR: 2.740, 95 % CI: 1.797–4.177, *p* < 0.001), LER (OR: 2.705, 95% CI: 1.752–4.176, *p* < 0.001), EMR (OR:1.654, 95% CI 1.123–2.436, *p* = 0.011), ELR (OR: 2.112, 95% CI 1.424–3.134, *p* < 0.001), and eosinophil count (OR: 1.833, 95% CI 1.276–2.635) were independent predictors for development of MACE. *Conclusions:* Patients with ADHF and reduced ejection fraction who developed MACE within the first six months of index hospitalization had lower levels of absolute eosinophil and lymphocyte counts, and EMR and ELR values, whereas NER and LER were higher compared to those without MACE. The eosinophil indices were independently associated with mortality and MACE development. The eosinophil indices may be used to estimate MACE likelihood with acceptable sensitivity and specificity.

## 1. Introduction

Heart failure (HF) is associated with substantial mortality and morbidity with an estimated prevalence of 1–2% in the adult population [1]. Currently, upwards of 64 million individuals are considered to live with HF (2017 data) [2]. A relatively recent meta-analysis including 60 studies with >1 million HF patients estimated 1-, 2-, 5- and 10-year survival rates of 87%, 73%, 57% and 35%, respectively [3]. In HF, need for hospitalization is among the independent risk factors of mortality regardless of left ventricular ejection fraction (LVEF) [4]. Advanced age, male sex, poor left ventricular systolic function, hyponatremia, low systolic blood pressure, ventricular arrhythmias, intraventricular conduction delays, low functional capacity, and renal dysfunction have also been shown to unfavorably influence survival in patients with HF [5].

Acute decompensation of heart failure defines a sudden worsening in HF symptoms often associated with a sudden increase in left ventricular filling pressure, causing volume accumulation in the lungs. Several simple, fast and readily available markers have been shown to predict mortality in subjects with heart failure with reduced ejection fraction (HFrEF). Particularly interesting is the fact that inflammatory markers may be associated with acute worsening and subsequent mortality risk, as demonstrated by studies showing that inflammation plays a critical role in the development and progression of HF [6]. For instance, the C-reactive protein-to-albumin ratio has been shown to be associated with advanced HF and poor hemodynamics in patients with reduced ejection fraction [7]. Mean platelet volume, leukocytes, and relative lymphocyte count are among the inflammation-related parameters that are increased in ADHF. Recently, elevation of Neutrophil-to-lymphocyte ratio (NLR), a simple and readily available marker of inflammation, was reported to be associated with unfavorable outcomes in patients with HF, regardless of left ventricular EF [8]. However, data concerning eosinophil indices, including neutrophil-to-eosinophil ratio (NER), leukocyte-to-eosinophil ratio (LER), eosinophil-to-lymphocyte ratio (ELR), and eosinophil-to-monocyte ratio (EMR) are limited.

Patients with HFrEF who present with ADHF are the most critical patients among those presenting with ADHF for which mortality is relatively high. Risk stratifying may help to identify those who need intensive management. With this study, we aimed to investigate the relationship between eosinophil indices and mortality and major adverse cardiovascular events (MACE) in ADHF patients with reduced EF.

## 2. Materials and Methods

A total of 395 consecutive patients admitted to the intensive care unit (ICU) with ADHF and reduced EF between January 2017 and December 2021 were enrolled in this retrospective study. HFrEF was defined as having an LVEF of ≤40% according to 2021 ESC guidelines for the diagnosis and treatment of acute and chronic heart failure. Demographic data were retrieved from patient charts and the institutional digital database. Patients with an LVEF > 40%, NYHA Class I and II subjects, those with advanced liver, kidney diseases or malignancies, patients in whom life expectancy was extremely short and patients with autoimmune, allergic, or infectious diseases that affect the eosinophil indices were not included in the study. Additionally, also, patients with concomitant acute ischemic events, who underwent coronary angiography, PCI or cardiac resynchronization therapy were excluded. The study was approved by the local ethics committee.

Conventional guidelines had been followed for the in-hospital treatment of all patients with ADHF and reduced EF, including loop diuretics, vasodilators, inotropes/vasopressors [9]. Blood tests including complete blood count, blood urea nitrogen and creatinine, electrolytes, liver enzymes, and NT-pro BNP were ordered upon admission to the ICU. All patients underwent transthoracic echocardiography. LVEF was measured with 2-dimensional echocardiography via the modified Simpson method [10]. Data concerning 6-month all-cause mortality and MACE were retrieved from the institutional digital database. The definition of MACE was the composite of total death and re-hospitalization for HF within 6 months of initial hospitalization.

The primary outcome measure of this study was all-cause mortality within the first 6 months of index hospitalization and its relationship with eosinophil indices, including NER, LER, ELR and EMR. The secondary outcome measure of this study was the assessment of MACE within the first 6 months of index hospitalization and its association with eosinophil indices.

### Statistical Analysis

All analyses were performed on SPSS v25 (IBM, Armonk, NY, USA). Histogram and Q-Q plots were evaluated to determine whether continuous variables were normally distributed. Numerical data with normal distribution were presented with mean ± standard deviation values, while numerical data with non-normal distribution were presented with median (25% and 75% percentile) values. Categorical data were presented as absolute and relative frequency (n, %). Between-groups analysis of numerical variables were performed with the independent samples t-test or the Mann–Whitney U test depending on normality of distribution. Between-groups analysis of categorical variables were performed with appropriate chi-square tests (Pearson, continuity correction). MACE prediction performance of the variables were assessed by using receiver operating characteristic (ROC) curve analysis. Optimal cut-off points were determined by the Youden index. Cox regression analyses were performed to determine prognostic factors independently associated with mortality and MACE. Variables demonstrating significance in univariate analysis were included into the multivariable Cox regression model. The threshold for statistical significance was accepted as *p* < 0.05.

## 3. Results

Longitudinal data was available for 395 subjects (mean age 76.51 ± 11.59 years, 56.5% male). MACE occurred in 176 (44.5%) subjects included in the study group. NT-pro BNP, body mass index, frequency of male sex, number of NYHA Class IV patients, blood urea nitrogen and creatinine, and NER and LER were significantly higher in subjects with MACE. Absolute eosinophil, lymphocyte and basophil count, hemoglobin, serum Na^+^, albumin, CRP, and EMR and ELR were significantly lower in subjects with MACE compared to those without (Table 1). Length of hospital stay was also longer in subjects developing MACE than those without MACE.

ROC curve analysis to predict 6-month mortality revealed that a cut-off value of 262 for NER (AUC: 0.699, 95% CI: 0.641–0.758, *p* < 0.001), 264.25 for LER (AUC: 0.693, 95% CI: 0.634–0.753, *p* < 0.001), 0.025 for absolute eosinophil count (AUC: 0.693, 95% CI: 0.632–0.754, *p* < 0.001), 0.064 for EMR (AUC: 0.613, 95% CI: 0.632–0.754, *p* < 0.001), 0.057 for ELR (AUC: 0.660, 95% CI: 0.597–0.723, *p* < 0.001), 7.785 for neutrophil (AUC: 0.637, 95% CI: 0.574–0.700, *p* < 0.001) and 0.84 for lymphocyte (AUC: 0.606, 95% CI: 0.535–0.677, *p* = 0.002) could be used with acceptable sensitivity and specificity (Table 2).

Kaplan–Meier survival analysis demonstrated that 6-month survival rate was significantly higher in patients with an NER value of ≤262 (Log rank: 35.59, *p* < 0.001), an LER of ≤264.25 (Log rank: 34.89, *p* < 0.001), an absolute eosinophil count of ≥0.025 (Log rank: 32.96, *p* < 0.001), an EMR of ≥0.064 (Log rank: 29.34, *p* < 0.001), and an ELR of ≥0.057 (Log rank: 24.61, *p* < 0.001) (Figure 1).

ROC curve analysis indicated that a cut-off value of 102.46 for NER (AUC: 0.699, 95% CI: 0.571–0.681, *p* < 0.001), 137.02 for LER (AUC: 0.616, 95% CI: 0.560–0.671, *p* < 0.001), 0.045 for absolute eosinophil count (AUC: 0.615, 95% CI: 0.559–0.670, *p* < 0.001), 0.134 for EMR (AUC: 0.586, 95% CI: 0.529–0.643, *p* = 0.003), 0.057 for ELR (AUC: 0.572, 95% CI: 0.515–0.628, *p* = 0.015) 5,26 for neutrophil (AUC: 0.590, 95% CI: 0.535–0.646, *p* = 0.001) and 0.89 for lymphocyte (AUC: 0.601, 95% CI: 0.575–0.687, *p* < 0.001) could be used to estimate the 6-month MACE development with acceptable sensitivity and specificity (Table 3). 

Kaplan–Meier survival analysis demonstrated that 6-month MACE-free survival was significantly higher in patients with an NER value of ≤102.46 (Log rank: 37.17, *p* < 0.001), an LER of ≤137.02 (Log rank: 35.18, *p* < 0.001), an absolute eosinophil count of ≥0.045 (Log rank: 24.88, *p* < 0.001), an EMR of ≥0.134 (Log rank: 16,26 *p* < 0.001) and those with an ELR of ≥0.057 (Log rank: 24.61, *p* < 0.001) (Figure 2).

Multivariable cox regression revealed that NT-pro BNP (OR: 1.682, 95% CI: 1.106–2.312, *p* = 0.001), Na^+^ (OR: 0.932, 95% CI: 0.897–0.969, *p* < 0.001), NER (OR: 2.740, 95 % CI: 1.797–4.177, *p* < 0.001), LER (OR: 2.705, 95% CI: 1.752–4.176, *p* < 0.001), EMR (OR:1.654, 95% CI 1.123–2.436, *p* = 0.011), ELR (OR: 2.112, 95% CI 1.424–3.134, *p* < 0.001), eosinophil count (OR: 1.833, 95% CI 1.276–2.635), having NYHA Class IV symptoms (OR: 1.124, 95% CI: 1.082–1.466, *p* = 0.001), and long hospitalization (OR: 1.097, 95% CI: 1.072–1.122, *p* < 0.001) were independent predictors associated with the development of MACE (Table 4).

## 4. Discussion

This study demonstrates that subjects with ADHF and reduced EF who developed MACE within the first six months of index hospitalization had lower absolute eosinophil count, lymphocyte count, EMR and ELR, while they had higher NER and LER compared to those without MACE. Moreover, NER, LER, EMR, ELR and absolute eosinophil count were independently associated with mortality. NER, LER, absolute eosinophil, neutrophil and lymphocyte count, EMR, and ELR could be used to estimate mortality likelihood in patients with ADHF with acceptable sensitivity and specificity.

HFrEF is of concern due to its morbidity and mortality outcomes. Studies conducted in the early 90′s showed a significant positive correlation between HF and the circulating levels of pro-inflammatory cytokines, particularly TNF-alfa [11]. Data accumulated from the experimental and clinical studies over the last 25 years confirmed the role of the immune system in the development and progression of both acute and chronic HF [12]. Myocardial injury promoted by ischemia, invading pathogens, and hemodynamic derangement leads to the activation of innate and adaptive immune systems [13,14,15]. Dysregulation of the inflammatory response in the acute phase of HF may lead to chronic inflammation, which is associated with left ventricular dysfunction and remodeling [16,17]. Activation of the innate immune system due to myocardial injury results in a non-specific and global response against the agent causing injury [12]. Immune response affects cardiomyocyte function and the course of interstitial fibrosis, there by influencing left ventricular performance. The circulating levels of pro-inflammatory cytokines has been shown to increase with the worsening of HF [18].

The ratio of eosinophil count to monocyte count, abbreviated as EMR, has been shown to predict long-term all-cause mortality following acute ST-elevation myocardial infarction [19]. A study by Yu et al. that enrolled 521 patients with acute ischemic stroke found that low EMR was independently associated with poor outcome [20]. Another study conducted by Chen and colleagues analyzed 280 acute ischemic stroke patients who received intravenous thrombolytic therapy and determined that lower EMR was independently associated with poor outcome and mortality [21]. A recent study involving 126 patients with ADHF has shown that absolute eosinophil count was lower in deceased patients compared to survivors at 1-year follow-up. Moreover, the authors reported a 4.4-fold increase in mortality risk among patients with an absolute eosinophil count of <0.02 × 10^9^/L compared to those with greater absolute eosinophil count [22]. In another study that retrospectively analyzed patients with ADHF, a positive correlation with EMR and cardiovascular death or re-hospitalization for HF was determined, indicating that lower EMR was associated with higher the risk for death and HF-related re-hospitalization [23].

Patients with HFrEF who present with ADHF are the most critical patients among those presenting with ADHF for which mortality is relatively high. Risk stratifying may help to identify those who need intensive management. We, therefore, selected this study group, not including HFpEF, whether eosinophil indices could provide additional data to better categorize patients with high risk for MACE.

Previous studies have primarily focused on absolute eosinophil count and EMR; however, we included the assessment of other indices associated with eosinophil count, such as ELR, LER, and NER. Our findings show that NER, LER, and ELR in addition to EMR are predictive for 6-month MACE in patients with ADHF. The changes in eosinophil count in early HF may be associated with myocardial injury and atherosclerosis; however, there is conflicting data regarding their possible protective or injurious effects with regard to circulatory levels and cardiac recruitment [24,25,26]. Traditionally, eosinophils are believed to possess a destructive role against several pathogens and contribute to the regulation of inflammation particularly in allergic diseases such as asthma. Eosinophils release immunosuppressive cytokines including IL-10, IL-4, and IL-13 [27]. However, novel data show that eosinophils are incorporated in several clinical conditions independent of parasite infection and allergy. Activated eosinophils are considered to promote angiogenesis by releasing endothelial growth factor and chemokines and cytokines [28]. Lack of eosinophils may be associated with inhibition of angiogenesis in subjects with coronary artery disease, and therefore, may contribute to the progression to HFrEF [29]. Eosinophilic activity also has been shown to contribute to fibroblast differentiation in damaged pulmonary tissue and release of proteases, and cytokines, which are critical components of tissue remodeling [30]. Although not specifically demonstrated in the myocardium, reportedly, there is a local increase (or recruitment) of proteases and cytokines during myocardial injury associated with ADHF [31] Moreover, there are also reports suggesting an inverse correlation between EMR and sympathoadrenal activity, which is a hallmark for the development of left ventricular remodeling –the critical pathophysiologic phenomenon responsible for LV dilatation and left ventricular dysfunction [22]. The decrease in eosinophil count during inflammatory processes results from cellular destruction in peripheral tissues, suppression of mature eosinophil migration from the bone marrow, accumulation of eosinophils in inflammatory sites, and bone marrow suppression [32]. Eosinopenia may also occur under acute stress conditions mediated by adrenal glucocorticoids and epinephrine [33]. Contrary to our findings, it should be considered that acute heart failure patients with reduced cardiac function may result in elevated eosinophils, as determined in the study of Rao et al. [34].

In this context, we suggest that eosinophil indices (NER, LER, and EMR), which are simple to calculate and readily available, may provide critical clues concerning the prognosis of patients with ADHF. The risk stratification can be determined by examining the CBC parameters after admission of ADHF with reduced EF cases to the ICU. Cases that are determined to be at higher risk for MACE can be followed up and treated by taking more serious precautions. Further, prospective studies with larger sample size will definitely be valuable to address the role of eosinophil indices in the estimation of HF prognosis and relationships with ADHF and MACE.

### Limitations

The retrospective design and relatively small size are the major drawbacks. A multivariate analysis based on so many variables has only limited statistical value and should have had a larger sample size. However, in its current form, this study provides important insight concerning the role of inflammation, and, in particular, eosinophils in the prognosis of ADHF. Information derived from our findings can promote further research addressing the role of eosinophil indices in HF.

## 5. Conclusions

Our data suggest that patients with ADHF who develop MACE within the first 6 months after index hospitalization have lower absolute eosinophil count, lymphocyte counts, EMR and ELR, whereas NER and LER are increased compared to those without MACE development. Furthermore, NER, LER, EMR, ELR and absolute eosinophil count were found to be independently associated with mortality and with the development of MACE in our group of ADHF patients with reduced EF. It appears that NER, LER, absolute eosinophil count, EMR, and ELR could be used to estimate MACE likelihood with acceptable sensitivity and specificity.

## Figures and Tables

**Figure 1 medicina-58-01455-f001:**
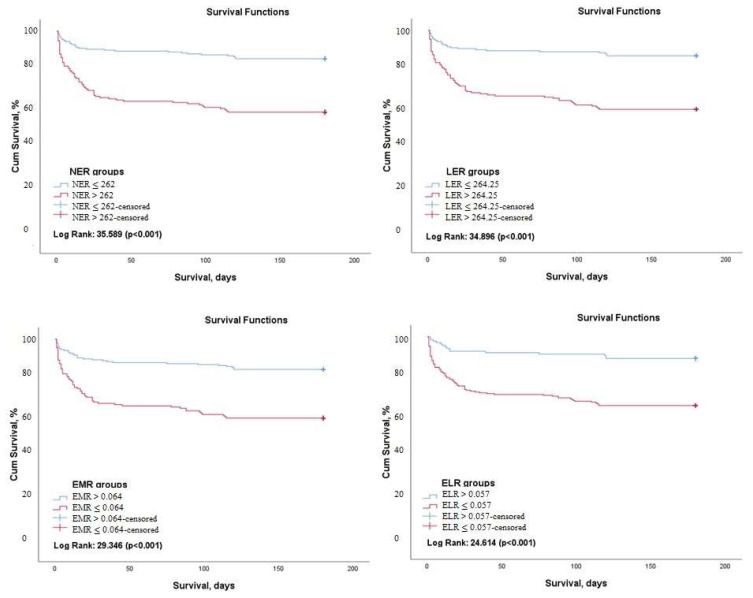
Kaplan–Meier survival analysis demonstrating 6 months mortality free survival according to NER, LER, EMR, and ELR (NER: Neutrophil-to-Eosinophil ratio, LER: Leukocyte-to-Eosinophil Ratio, EMR: Eosinophil-to-Monocyte Ratio, ELR: Eosinophil-to-Lymphocyte ratio).

**Figure 2 medicina-58-01455-f002:**
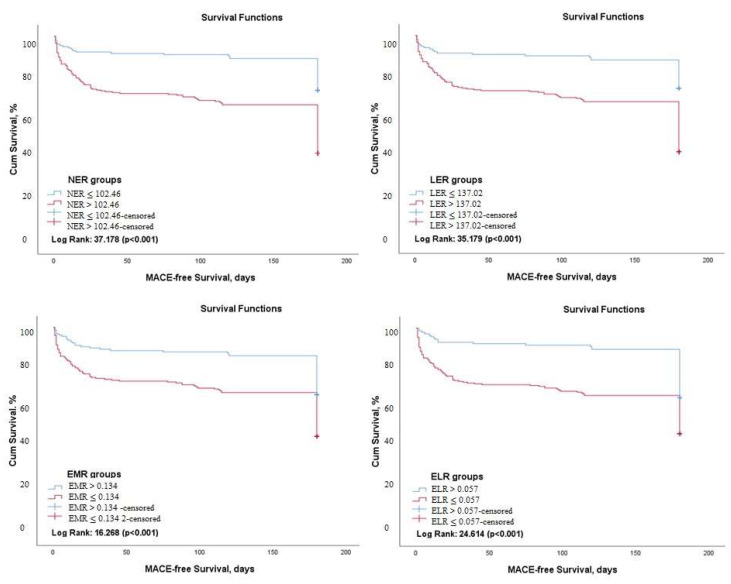
Kaplan–Meier survival analysis demonstrating 6-month MACE-free survival according to NER, LER, EMR, and ELR (NER: Neutrophil-to-Eosinophil ratio, LER: Leukocyte-to-Eosinophil Ratio, EMR: Eosinophil-to-Monocyte Ratio, ELR: Eosinophil-to-Lymphocyte ratio).

**Table 1 medicina-58-01455-t001:** Comparison of baseline demographic, clinical, laboratory and echocardiographic data of all study patients, MACE (+) and MACE (−) patient groups.

	All Patients (n = 395)	MACE (+) (n = 176)	MACE (−) (n = 219)	*p*-Value
Age (years)	76.5 ± 11.6	76.2 ± 10	76.8 ± 12.8	0.650
Male sex, n (%)	223 (%56.5)	82 (%46.5)	141 (%64.3)	0.001
BMI (kg/m^2^)	25.2 ± 5.4	23.4 ± 4.9	26.1 ± 5.3	0.007
NYHA, n (%)				0.010
Class III	217 (%54.9)	62 (%35.2)	155 (%70.7)	
Class IV	178 (%45.1)	114 (%64.7)	64 (%29.2)	
AF, n (%)	227 (%57.5)	96 (%54.5)	131 (%59.8)	0.292
DM, n (%)	158 (%40)	79 (%44.8)	79 (%36.1)	0.076
HT, n (%)	324 (%82)	134 (%76.1)	190 (%86.7)	0.006
CAD, n (%)	287 (%72.7)	122 (%69.3)	165 (%75.3)	0.182
LoS (days)	7 (4–11)	11(6–19)	5 (3–8)	<0.001
Ejection fraction (%)	31.8 ± 8.5	31.1 ± 7.9	32.4 ± 8.8	0.124
sPAP (mmHg)	48.6 ± 11.0	49.4 ± 10.9	48.0 ± 11.1	0.365
Leukocytes (10^9^/L)	9.3 ± 3.4	9.4 ± 3.1	9.2 ± 3.7	0.654
Neutrophil (10^9^/L)	7.4 ± 3.3	7.7 ± 3.0	7.1 ± 3.5	0.050
Lymphocyte (10^9^/L)	1.3 ± 0.8	1.1 ± 0.6	1.4 ± 0.9	<0.001
Monocyte (10^9^/L)	0.56 (0.25–0.70)	0.77 (0.36–1.72)	0.59 (0.23–0.80)	0.078
Eosinophil (10^9^/L)	0.08 ± 0.02	0.06 ± 0.01	0.10 ± 0.04	0.002
Basophil (10^9^/L)	0.03 ± 0.03	0.03 ± 0.02	0.04 ± 0.03	<0.001
Hemoglobin (g/dL)	12.00 ± 2.01	11.76 ± 2.04	12.19 ± 1.98	0.039
Hematocrit (%)	37.3 ± 5.9	36.8 ± 6.1	37.6 ± 5.7	0.190
MCV (fL)	88.1 ± 7.5	86.5 ± 7.7	89.4 ± 7.1	<0.001
Platelets (10^9^/L)	235.9 ± 88.6	243.0 ± 98.8	230.2 ± 81.2	0.158
MPV (fL)	9.9 ± 1.2	9.8 ± 1.2	9.9 ± 1.3	0.195
Glucose (mg/dL)	160.4 ± 41.0	159.3 ± 48.3	161.4 ± 44.9	0.800
BUN (mg/dL)	39.5 ± 12.1	45.0 ± 14.2	35.1 ± 11.2	<0.001
ALT (u/L)	21 (12–37)	51 (24–101)	69 (29–119)	0.165
AST (u/L)	25 (21–51)	68 (56–126)	59 (47–118)	0.366
Potassium (mmol/L)	5.0 ± 1.5	5.3 ± 1.7	4.7 ± 0.6	0.196
Sodium (mmol/L)	137.5 ± 5.6	136.5 ± 6.5	138.2 ± 4.5	0.001
CRP (mg/L)	12(4.2–25. 8)	18 (6–39.2)	26 (12.9–51.7)	0.016
Albumin (g/dL)	3.8 ± 0.5	3.7 ± 0.5	3.8 ± 0.5	0.004
Creatinine (mg/dL)	1.5 ± 0.6	1.6 ± 0.7	1.4 ± 0.5	0.002
NT-proBNP (×10^3^), (pg/mL)	5.7 (2.9–13.1)	9.8 (4.4–14.3)	4.6 (1.9–12.7)	0.011
Mortality, n (%)	96 (%24.3)	-	-	-
Re-hospitalization, n (%)	114 (%28.9)	-	-	-
MACE, n (%)	176 (%44.6)	-	-	-
NER	984.1 ± 270.1	1527.4 ± 275.7	547.5 ± 108.6	<0.001
LER	1147.0 ± 335.4	1755.9 ± 444.0	657.6 ± 124.4	<0.001
EMR	0.19 ± 0.05	0.17 ± 0.04	0.20 ± 0.05	0.001
ELR	0.07 ± 0.02	0.06 ± 0.02	0.08 ± 0.01	0.009

Numerical data with normal distribution were presented as mean ± standard deviation, and numeric data with non-normal distribution as median (25% and 75% percentile) values. MACE: Major Adverse Cardiovascular Events, BMI: Body Mass Index, NYHA: New York Heart Society, AF: Atrial Fibrillation, DM: Diabetes Mellitus, HT: Hypertension, CAD: Coronary Artery Disease, MCV: Mean Corpuscular Volume, MPV: Mean Platelet Volume, BUN: Blood Urea Nitrogen, ALT: Alanine aminotransferase, AST: Aspartate aminotransferase, CRP: C-reactive protein, NER: Neutrophil-to-Eosinophil Ratio, LER: Leukocyte-to-Eosinophil Ratio, EMR: Eosinophil-to-Monocyte Ratio, ELR: Eosinophil-to-Lymphocyte Ratio.

**Table 2 medicina-58-01455-t002:** ROC curve analysis demonstrating the cut-off values for eosinophil indices to predict the 6-moths mortality.

	AUC	95%CI	*p*-Value	Sensitivity	Specificity	Cut-Off
NER	0.699	0.641–0.758	<0.001	69%	64%	262.00
LER	0.693	0.634–0.753	<0.001	74%	60%	264.25
Eosinophil count	0.693	0.632–0.754	<0.001	65%	66%	0.025
EMR	0.663	0.602–0.775	<0.001	66%	64%	0.064
ELR	0.660	0.597–0.723	<0.001	84%	45%	0.057
Neutrophil	0.637	0.574–0.700	<0.001	57%	69%	7.785
Lymphocyte	0.606	0.535–0.677	0.002	45%	79%	0.84
Monocyte	0.538	0.472–0.604	0.264	58%	53%	0.555

AUC: Area under the curve, NER: Neutrophil-to-Eosinophil ratio, LER: Leukocyte-to-Eosinophil ratio, EMR: Eosinophil-to-Monocyte ratio, ELR: Eosinophil-to-Lymphocyte ratio.

**Table 3 medicina-58-01455-t003:** ROC curve analysis demonstrating the cut-off values for eosinophil indices to predict the 6-moths MACE.

	AUC	95%CI	*p*-Value	Sensitivity	Specificity	Cut-Off
NER	0.626	0.571–0.681	<0.001	76%	53%	102.46
LER	0.616	0.560–0.671	<0.001	77%	51%	137.02
Eosinophil count	0.615	0.559–0.670	<0.001	67%	55%	0.045
EMR	0.586	0.529–0.643	0.003	70%	49%	0.134
ELR	0.572	0.515–0.628	0.015	72%	46%	0.057
Neutrophil	0.590	0.535–0.646	0.002	81%	40%	5.26
Lymphocyte	0.601	0.575–0.687	<0.001	42%	82%	0.89
Monocyte	0.566	0.508–0.623	0.025	35%	77%	0.405

AUC: Area under the curve, NER: Neutrophil-to-Eosinophil ratio, LER: Leukocyte-to-Eosinophil Ratio, EMR: Eosinophil-to-Monocyte Ratio, ELR: Eosinophil-to-Lymphocyte ratio.

**Table 4 medicina-58-01455-t004:** Results of univariate and multivariable Cox regression analysis for the determination of mortality and MACE independent predictors.

	Mortality				MACE			
	Univariate		Multivariable		Univariate		Multivariable	
	HR (%95 CI)	*p*-Value	HR (%95 CI)	*p*-Value	HR (%95 CI)	*p*-Value	HR (%95 CI)	*p*-Value
NER	3.455 (2.233–5.344)	<0.001	3.509 (1.956–6.296)	<0.001 *	2.657 (1.873–3.770)	<0.001	2.740 (1.797–4.177)	<0.001 **
LER	3.593 (2.277–5.671)	<0.001	3.587 (1.969–6.535)	<0.001 *	2.633 (1.844–3.758)	<0.001	2.705 (1.752–4.176)	<0.001 **
EMR	3.034 (1.984–4.641)	<0.001	2.846 (1.641–4.936)	<0.001 *	1.916 (1.386–2.649)	<0.001	1.654 (1.123–2.436)	0.011 **
ELR	3.652 (2.104–6.337)	<0.001	4.302 (2.064–8.968)	<0.001 *	1.871 (1.345–2.603)	<0.001	2.112 (1.424–3.134)	<0.001 **
Eosinophil	3.197 (2.097–4.873)	<0.001	3.379 (1.916–5.959)	<0.001 *	2.091 (1.525–2.869)	<0.001	1.833 (1.276–2.635)	0.001 **
Neutrophil	1.099 (1.043–1.159)	<0.001	1.077 (1.001–1.158)	0.046 *	1.054 (1.012–1.099)	0.012	1.053 (0.998–1.112)	0.061
Lymphocyte	0.671 (0.471–0.957)	0.028	0.726 (0.454–1.161)	0.181	0.600 (0.454–0.794)	<0.001	0.822 (0.598–1.129)	0.226
Monocyte	0.687 (0.371–1.271)	0.232			1.047 (0.942–1.164)	0.391		
Creatinine	1.566 (1.162–2.109)	0.003	0.763 (0.494–1.177)	0.221	1.483 (1.176–1.870)	0.001	0.982 (0.719–1.342)	0.909
Albumin	0.313 (0.193–0.508)	<0.001	0.572 (0.331–0.986)	0.044	0.561 (0.401–0.786)	0.001	0.757 (0.521–1.102)	0.146
CRP	1.001 (0.994–1.007)	0.797			0.995 (0.989–1.001)	0.082	0.989 (0.980–1.008)	0.116
Sodium	0.919 (0.891–0.948)	<0.001	0.923 (0.884–0.964)	<0.001	0.940 (0.915–0.966)	<0.001	0.932 (0.897–0.969)	<0.001
Hemoglobin	0.860 (0.771–0.960)	0.007	0.896 (0.775–1.036)	0.139	0.909 (0.839–0.985)	0.020	0.946 (0.852–1.050)	0.298
Hypertension	0.462 (0.299–0.714)	0.001	0.382 (0.218–0.670)	0.001	0.584 (0.413–0.826)	0.002	0.630 (0.403–1.005)	0.063
EF	0.993 (0.970–1.017)	0.571			0.988 (0.971–1.006)	0.194		
Age	1.001 (0.984–1.018)	0.904			0.998 (0.985–1.010)	0.785		
Hospitalization Period	1.026 (0.994–1.059)	0.116			1.087 (1.066–1.109)	<0.001	1.097 (1.072–1.122)	<0.001
NYHA class IV	1.201 (1.009–2.208)	0.002	1.044 (1.002–1.206)	0.011	1.331 (1.018–2.266)	0.001	1.124 (1.082–1.466)	0.001
NT-pro-BNP	1.568 (1.124–2.804)	0.004	1.502 (1.126–2.101)	0.007	1.688 (1.104–2.866)	0.002	1.682 (1.106–2.312)	0.001

* Multivariable modeling was performed with creatinine, albumin, sodium, hemoglobin, neutrophil, hypertension, NYHA class IV and NT-pro-BNP variables. ** Multivariable modeling was performed with creatinine, albumin, CRP, sodium, hemoglobin, hypertension, length of stay, NHYA class IV and NT-pro-BNP variables. CRP: C-reactive protein, EF: Ejection fraction, ELR: Eosinophil-to-Lymphocyte ratio, EMR: Eosinophil-to-Monocyte Ratio, LER: Leukocyte-to-Eosinophil Ratio, MACE: Major Adverse Cardiovascular Events, NER: Neutrophil-to-Eosinophil ratio, NYHA: New York Heart Society.

## Data Availability

Data are available on request due to privacy.

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
