# Peer review of "The Predictive Value of Eosinophil Indices for Major Cardiovascular Events in Patients with Acute Decompensated HFrEF"

_medicina, 2022, doi:10.3390/medicina58101455_

Round 1
Reviewer 1 Report
Colleagues Vural an Aydin report in their article "The Predictive Value of Eosinophil Indices for Major Cardiovascular Events in Patient with Acute Decompensatet HFref" about eosinophil indices and their influence on MACE and disease progression. They conclude, that lower eosinophil and lymphocyte counts as well as EMR and ELR were associated with higher rates of MACE within the first 6 months after a first index episode of acute decompensated heart failure. The manuscript is well written, but a there are few comments, that need to be addressed.
1) Abstract and Methods: Please report number of patients, that were included in the analysis. Why were patients with EF > 40% excluded? Please comment. Is there any information on causes for HFref? Ischemic?
2) Statistics: Which variables were included in the multivariate Cox? If all variables are included as seen in table 4, you should discuss the limitation, that such regression analysis is unstable because of the relation of variables and N. Especially by taking NER and EMR or LER and ELR into the analysis you overweight eosinophiles and lymphozytes as they are part of two variables. It would be interesting how this analysis results with only eosinophiles and lymphozytes - especially in relation to the other variables. Please comment on that on provide this analysis for me.
3) Figures: Please optimize the Figures, as these figures are direct SPSS output. P-values can not be seen, remove grid lines, axis labels should corrected: X-axis (days??), y-axis (%)
4) As you discuss ischemic causes: How many patients received coronary angiography and even PCI? Would this differention of patients lead to different results? Treated ischemia vs. non-ischemic patients?
What are your clinicial implications of your findings? Do your findings result in different treatment of patients?
Author Response
Dear Reviewer,
We would like to thank you for insightful comments and suggestions. We made all possible changes that were suggested.
1) Abstract and Methods: Please report number of patients, that were included in the analysis. Why were patients with EF > 40% excluded? Please comment. Is there any information on causes for HFref? Ischemic?
Response:
-The number of patients added to the abstract and materials-methods section.
-In our study, we examined patients with HFrEF. According to 2021 ESC guidelines for the diagnosis and treatment of acute and chronic heart failure, HFrEF was defined as having an LVEF of < 40 %. We added reference guidelines to the materials and methods section.
-We actually have data on the etiology of HFrEF. In Table 1, instead of coronary artery disease, HR was written inadvertently. Sorry for our carelessness. We made corrections in table 1.
2) Statistics: Which variables were included in the multivariate Cox? If all variables are included as seen in table 4, you should discuss the limitation, that such regression analysis is unstable because of the relation of variables and N. Especially by taking NER and EMR or LER and ELR into the analysis you overweight eosinophiles and lymphozytes as they are part of two variables. It would be interesting how this analysis results with only eosinophiles and lymphozytes - especially in relation to the other variables. Please comment on that on provide this analysis for me.
Response: Not all variables are included in the multivariate analysis, as stated ‘Cox regression analyses were performed to determine prognostic factors independently associated with mortality and MACE. Variables demonstrating significance in univariate analysis were included into the multivariable cox regression model. The threshold for statistical significance was accepted as p<0.05.’ in the statistical analysis, only variables are demonstrating significance ( p<0.05 ) in the univariate analysis were included into multivariate analysis both mortality and MACE.
Eosinophil count in univariate and multivariate analysis for mortality and MACE was included in our article. Since this was not the focus of our study, we did not include lymphocyte, monocytes, and neutrophil counts in the univariate and multivariate analysis. We have added these analyzes upon request. But their values alone were not as strong as the eosinophilic indices in which they were included. And also, the variables alone were mentioned at ROC analysis
3) Figures: Please optimize the Figures, as these figures are direct SPSS output. P-values can not be seen, remove gridlines, axis labels should corrected: X-axis (days??), y-axis (%)
Response: The figures were corrected according to the suggestions.
4) As you discuss ischemic causes: How many patients received coronary angiography and even PCI? Would this differention of patients lead to different results? Treated ischemia vs. non-ischemic patients?
Response: Since revascularization could affect the results, patients with acute coronary syndrome and the possibility of reversible ischemia, or who underwent coronary angiography and PCI were excluded. Similarly, patients received cardiac resynchronization therapy were excluded. Patients with ischemic or non-ischemic heart failure were treated similarly. No treatment specifically aimed at reducing ischemia was applied. ‘And also, patients with concomitant acute ischemic events, who underwent coronary angiography, PCI or cardiac resynchronization therapy were excluded’ added to methods section.
What are your clinicial implications of your findings? Do your findings result in different treatment of patients?
Response: ‘The risk stratification can be determined by examining the CBC parameters after admission of ADHF with reduced EF cases to the ICU. Cases that are determined to be at higher risk for MACE can be followed up and treated by taking more serious precautions. ‘ We added to last paragraph of discussion )
Reviewer 2 Report
This manuscript focuses on important eosinophil indicators in the prediction of cardiovascular events in patients with acute heart failure. This article focuses on the ratio of eosinophils to neutrophils, leukocytes, and lymphocytes in patients with acute heart failure and points to their prognostic relevance. Eosinophils have been implicated as a potential cause of myocardial injury of widely varying severity, ranging from acute myocarditis to endocardial fibrosis. In general, eosinophils play two roles in the immune system. 1) Eosinophils are able to consume foreign substances. 2) Eosinophils promote inflammation and play a beneficial role in sequestering and controlling disease sites. Sometimes, however, inflammation can be greater than necessary, leading to bothersome symptoms and tissue damage.
This paper is very interesting, but I believe that the reader's understanding would be improved if the following issues were addressed.
Major
1) In this case, patients with HFrEF were included, but on the summary, they are ADHF. The point about not including HFpEF patients in the discussion should be explained. The possibility that acute heart failure in patients with reduced cardiac function may result in elevated eosinophils should be considered.
2) Why were eosinophils significantly lower in patients with MACE (+) compared to those with MACE (-)? In addition to that, you should explain why you use ratios such as LER, EMR, ELR, etc.
Minor
1) The paper should be changed to significant figures (e.g., to one decimal place) to make it easier to read. For example, mean age 76.5 ± 11.6 years etc.
Author Response
Dear Reviewer,
We would like to thank you for insightful comments and suggestions. We made all possible changes that were suggested.
Major
- In this case, patients with HFrEF were included, but on the summary, they are ADHF. The point about not including HFpEF patients in the discussion should be explained. The possibility that acute heart failure in patients with reduced cardiac function may result in elevated eosinophils should be considered.
Response: ‘Patients with HFrEF who present with ADHF are the most critical patients among those presenting with ADHF for which mortality is relatively high. Risk stratifying may help to identify those who need intensive management. We, therefore, selected this study group, not including HFpEF, whether eosinophil indices could provide additional data to better categorize patients with high risk for MACE.’ And ‘Contrary to our findings, it should be considered that acute heart failure patients with reduced cardiac function may result in elevated eosinophils, as determined in the study of Rao et al. [32]’ added to discussion.
- Why were eosinophils significantly lower in patients with MACE (+) compared to those with MACE (-) ? In addition to that, you should explain why you use ratios such as LER, EMR, ELR, etc.,
Response: In our article, the relationship between the prognosis of acute heart failure and the inflammatory process was mentioned and thus we think that the inflammatory response is higher in MACE+ patients. ‘The decrease in eosinophil count during inflammatory processes results from cellular destruction in peripheral tissues, suppression of mature eosinophil migration from the bone marrow, accumulation of eosinophils in inflammatory sites, and bone marrow suppression. Eosinopenia may also occur under acute stress conditions mediated by adrenal glucocorticoids and epinephrine.’ added to discussion.
With statistical data added in our revised article, we have shown that the ratios (LER, EMR, ELR,etc.) have stronger statistically values than their values alone (eosinophils, lymphocytes, neutrophyl and monocytes ) We used these ratios because they have higher predictive values. (Please review revised tables 2, 3 and 4)
Minor
1) The paper should be changed to significant figures (e.g., to one decimal place) to make it easier to read. Fo rexample, mean age 76.5 ± 11.6 yearsetc.
Response: The paper was changed as suggested as possible as, one or two decimal places.
Round 2
Reviewer 1 Report
Thank you for corresponding to my suggestions. Nevertheless there are minor issues, that need to be addressed:
1) Figures: Scaling y-axis schould be scaled x10, as mortality in percent is displayed. commas should be dots (e.g. 0,8 should be 80 or 80.0), also in the figure description (EMR/groups: EMR >0,134 should be EMR >0.134 etc.)
2) Limitations: Authors should add the limitation, that a multivariate analysis based on such many variables has only limited statistical value and bigger sample sizes would be desirable.
Author Response
Dear Reviewer,
Thank you for the opportunity to revise our manuscript. We have revised the manuscript according to suggestions
1) Figures: Scaling y-axis schould be scaled x10, as mortality in percent is displayed. commas should be dots (e.g. 0,8 should be 80 or 80.0), also in the figure description (EMR/groups: EMR >0,134 should be EMR >0.134 etc.)
Response: Figures was edited according to suggestions
2) Limitations: Authors should add the limitation, that a multivariate analysis based on such many variables has only limited statistical value and bigger sample sizes would be desirable.
Response: 'A multivariate analysis based on so many variables has only limited statistical value and should have had a larger sample size' added to limitations
